# Research on the Method of Foreign Object Detection for Railway Tracks Based on Deep Learning

**DOI:** 10.3390/s24144483

**Published:** 2024-07-11

**Authors:** Shanping Ning, Feng Ding, Bangbang Chen

**Affiliations:** 1School of Mechatronic Engineering, Xi’an Technological University, Xi’an 710016, China; ningshanping@163.com (S.N.); chenbangbang@st.xatu.edu.cn (B.C.); 2Railway Transportation Institute, Guangdong Communication Polytechnic, Guangzhou 510650, China

**Keywords:** foreign object intrusion, detection method, semantic segmentation network, modified YOLOv5s, attention mechanism, loss function

## Abstract

Addressing the limitations of current railway track foreign object detection techniques, which suffer from inadequate real-time performance and diminished accuracy in detecting small objects, this paper introduces an innovative vision-based perception methodology harnessing the power of deep learning. Central to this approach is the construction of a railway boundary model utilizing a sophisticated track detection method, along with an enhanced UNet semantic segmentation network to achieve autonomous segmentation of diverse track categories. By employing equal interval division and row-by-row traversal, critical track feature points are precisely extracted, and the track linear equation is derived through the least squares method, thus establishing an accurate railway boundary model. We optimized the YOLOv5s detection model in four aspects: incorporating the SE attention mechanism into the Neck network layer to enhance the model’s feature extraction capabilities, adding a prediction layer to improve the detection performance for small objects, proposing a linear size scaling method to obtain suitable anchor boxes, and utilizing Inner-IoU to refine the boundary regression loss function, thereby increasing the positioning accuracy of the bounding boxes. We conducted a detection accuracy validation for railway track foreign object intrusion using a self-constructed image dataset. The results indicate that the proposed semantic segmentation model achieved an MIoU of 91.8%, representing a 3.9% improvement over the previous model, effectively segmenting railway tracks. Additionally, the optimized detection model could effectively detect foreign object intrusions on the tracks, reducing missed and false alarms and achieving a 7.4% increase in the mean average precision (IoU = 0.5) compared to the original YOLOv5s model. The model exhibits strong generalization capabilities in scenarios involving small objects. This proposed approach represents an effective exploration of deep learning techniques for railway track foreign object intrusion detection, suitable for use in complex environments to ensure the operational safety of rail lines.

## 1. Introduction

The technology of fully automatic train operation has become a hot topic of current research, and unmanned driving is even the goal of future railway operation [1]. However, the intrusion of foreign objects such as pedestrians, falling rocks, animals, etc., into railway lines poses a serious threat to the safety along the railway. Therefore, effectively identifying and detecting the intrusion of foreign objects on railway tracks is of great significance to the safe operation of the railway [2,3]. Existing detection methods for foreign object incursion can be divided into two categories: contact and non-contact. The contact detection method realizes direct contact perception of foreign objects by installing contact sensors, such as vibration optical fibers [4,5] and pulse electronic fences [6], on the protective net. However, this detection method has high installation costs and cannot detect objects that do not come into contact with the sensors [7]. Non-contact detection methods typically utilize non-contact sensors such as images and radars to perceive foreign objects. When detecting foreign object incursion, they can promptly issue warning signals to drivers [8,9]. In recent years, artificial intelligence, especially deep learning, has rapidly emerged and made significant progress in computer vision detection tasks such as image segmentation and object detection [10,11,12,13]. As an important branch of deep learning, convolutional neural networks provide new solutions for foreign object recognition due to their excellent detection performance, low-cost equipment, strong flexibility, and ease of deployment [14].

In railway scenarios, there are two types of vision-based detection equipment: those installed along the trackside and onboard equipment [15,16]. Equipment installed along the trackside can be used to monitor fixed scenarios, and the installation of this equipment involves a large number of pieces and complexity. However, cameras mounted on onboard equipment can detect various scenes along the trackside during train operation. Therefore, detecting foreign object incursion on railway tracks based on onboard equipment’s image data is both practical and economical. With the successful application of deep learning-based methods in the field of computer vision, railway track foreign object detection technology has made significant progress [17]. T. Ye et al. [18] proposed a stable lightweight feature extraction and adaptive feature fusion network for small object detection, aiming to detect obstacles in real-time railway traffic scenarios to ensure driving safety. Z. Nan et al. [19] developed a mechanical 3D LiDAR system for small objects, capable of monitoring obstacles with dimensions of 15 cm × 15 cm × 15 cm, providing a promising solution for railway transportation safety. S. Wang et al. [20] proposed a high-precision small object detection and recognition algorithm for small foreign objects in complex environments. By optimizing the backbone network, convolutional layers, and loss function of YOLOv5s, the network can focus more on small objects. These studies have optimized the deep learning-based object detection models in various aspects and achieved good recognition accuracy in foreign object detection. However, they only consider detecting all objects in the scene and have difficulty determining whether the objects are in dangerous areas. Z. Yu et al. [21] proposed the DALNet network for track detection, which dynamically constructs appropriate anchor lines for each rail instance. The dynamically generated anchor lines can more accurately locate the rails. Y. Weng et al. [22] presented an improved d-linknet convolutional neural network that integrates a specifically designed edge detection module to fuse multi-level features, thereby enhancing the model’s segmentation and extraction of target edges. While these methods have extracted specific detection areas to some extent, track extraction still needs to be strengthened under background interference, and the detection areas have not been divided into restricted zones.

When foreign objects appear in railway monitoring systems, timely and accurate detection and identification of these objects are crucial. Currently, semantic segmentation technology based on deep learning can achieve pixel-level segmentation of images [23] and has been widely applied in areas such as drivable regions [24]. Meanwhile, YOLO, as a high-performance real-time object detection model, can effectively detect foreign objects within the railway boundary area [25]. Therefore, these methods provide good theoretical and technical support for foreign object detection and segmentation of specific detection areas in railway monitoring systems. Combining the aforementioned issues and methods, this article proposes research on foreign object intrusion detection for railway tracks based on deep learning to detect and warn of foreign object intrusions on railway tracks. The main contributions of this study can be summarized as follows:The railway lane detection method has been improved to extract the entire track area of interest for this paper, establishing a track boundary model that provides important support for the division of dangerous areas on the track.The YOLOv5s model has been improved for the detection of small objects at long distances, improving cases of missed detections and false alarms and enhancing the detection accuracy and positioning accuracy of the model.Extensive experimental evaluations have been conducted on the proposed method, using multiple performance metrics for comprehensive assessment and comparison. Meanwhile, self-built railway foreign object incursion data has been used for recognition and incursion behavior judgment to verify the practicality of the proposed method in real-world scenarios.

The following sections of this paper are organized as follows: Section 2 introduces the experimental methods, including railway track detection, the establishment of a foreign object incursion model, foreign object detection, and incursion judgment. Section 3 presents the experimental results and demonstrates the effectiveness of the proposed method through comparisons. Section 4 summarizes the achievements of this paper and proposes directions for future work.

## 2. Foreign Object Incursion Detection Method

The intrusion of foreign objects along railway tracks seriously affects transportation safety, potentially leading to transportation disruptions, and even train derailments, resulting in severe safety accidents [26]. To effectively address this issue, machine vision technology can be employed for the detection of foreign object incursions on railway tracks. Semantic segmentation can extract the track structure to accurately describe the area of intrusion [27], while object detection technology can accurately identify foreign objects on the track surface [28]. The effective combination of these two technologies can effectively detect foreign object incursions on railway tracks. The specific steps are as follows: first, use a semantic segmentation model to extract the railway track and accurately describe the structural information of the track; second, determine the linear equation of the track using the least squares method to establish a track boundary model; and, finally, utilize object detection technology to detect foreign objects, judge the behavior of foreign object incursions, and take timely measures to ensure the safety of railway transportation.

By integrating the aforementioned issues and analysis, the behavior of foreign objects entering the restricted area is determined by utilizing the output results of semantic segmentation and object detection. The specific process is illustrated in Figure 1.

### 2.1. Establishment of Track Clearance Model

#### 2.1.1. Method of Track Extraction

Semantic segmentation refers to the processing of images or videos to accurately separate track information from the images [29]. Therefore, the segmentation method is an important technical foundation for realizing the detection of foreign object invasion. However, it faces challenges in terms of real-time performance and accuracy. When equipment analyzes videos, real-time and continuous calculations are required, hence the need for an efficient segmentation algorithm. In response to the aforementioned issues, considering that railway tracks have continuous, smooth, and structurally simple characteristics compared to road lanes, the widely used UNet network efficient semantic segmentation model is adopted. The overall flowchart of the segmentation method is shown in Figure 2.

As illustrated in Figure 3, the improved UNet network structure and processing flow are fully symmetrical in terms of the encoder and decoder. The encoder is designed to capture the key information of the image at different scales, resulting in five distinct feature maps after four convolution operations. The decoder is responsible for restoring the original image through precise information fusion with feature maps of corresponding scales, enabling the reuse of shallow features. Prior to the fusion of shallow features, an attention mechanism module is introduced to reconcile the information discrepancies between deep and shallow features. The prediction area leverages the final feature map to forecast the results, determining the category of each pixel.

The UNet network has been improved as follows: (a) As shown in Figure 4, before the shallow feature fusion, a channel and spatial attention module CBAM is added to the shallow features to reduce the information gap between deep and shallow features while enhancing the correlation of feature information in both spatial and channel dimensions; (b) the activation function in the encoder and decoder sections is Leaky ReLU, which effectively addresses the issue of having half of the gradients equal to 0.

#### 2.1.2. Feature Extraction

The image is divided equally on the *Y*-axis at intervals of 25 pixels, as shown in Figure 5a. Along a dividing line, the pixels of the background and the rails exhibit a distribution of “background + left rail + background + right rail”, which translates to a “0-1-0-1” arrangement in the output matrix. By iterating through each dividing line, the coordinate information of the pixel points on the left and right rails is calculated, and their average values are obtained to yield the characteristic coordinates of the left and right rails, as depicted in Figure 5b.

#### 2.1.3. Track Line Fitting

Based on the type of different tracks, the method shown in Figure 5 is adopted to calculate the linear or nonlinear distribution of track feature points. The least squares polynomial is used for track line fitting, and the process is as follows:

Given the coordinates $\left(x_{i},y_{i}\right)$ of the track feature points, the curve equation can be expressed as follows:(1)$$H(x)=a_{n}x^{n}+a_{n-1}x^{n-1}+....+a_{1}x+b$$

Expressing Equation (1) in matrix form gives the following:(2)H(x)=Xa

In the equation, *H* is an *m* × 1 vector representing the predicted values of the model, a is an *n* × 1 vector representing the coefficients to be solved in the equation, and *X* is an *m* × *n* dimensional matrix. The sum of squared errors can be expressed as follows:(3)$$R^{2}={\left\|H-Y\right\|}^{2}={\left\|Xa-Y\right\|}^{2}=(Xa-Y{)}^{T}(Xa\;-\;Y)$$

In the equation, Y represents the output matrix of the track pixel points, with a being the dimension of *m* × 1. Taking the derivative with respect to the coefficient a, we obtain the following:(4)$$\frac{\partial R^{2}}{\partial a}=2X^{T}Xa-2X^{T}Y=0$$

Solve for coefficient $a=(X^{T}X{)}^{-1}X^{T}Y$. By fitting the image in Figure 5, the fitting result is shown in Figure 6.

Following the above track-fitting steps, the linear equation for Figure 6 can be obtained as follows:(5)$$\left\{\begin{array}{c} H_{L}(x)=2.5x-290(116\leq x\leq 235)\\ H_{R}(x)=-2.5x+975(260\leq x\leq 390) \end{array}\right.$$

#### 2.1.4. Track Boundary Model

Based on the standard gauge of 1435 mm and the track boundary range of 4880 mm specified in the Chinese national standard GB-163.2020 [30], taking the straight track as an example (the method for curved tracks is the same), the schematic diagram of the mathematical model for determining the track boundary is shown in Figure 7a, which can be used to judge the behavior of foreign object intrusion, as illustrated in Figure 7b.

### 2.2. Foreign Object Detection

#### 2.2.1. The Improved YOLOv5s Model

YOLOv5s, as the smallest model in the YOLOv5 series, comes with five different configurations of n, s, m, l, and x to adapt to different scenarios [31]. Considering the need for real-time processing of railway boundary videos, YOLOv5s is chosen as the base model. This model consists of four core modules: input, backbone network, neck network, and prediction end. The input end utilizes Mosaic technology to enhance the background, combined with adaptive anchor boxes and image scaling techniques to optimize data input. The backbone network integrates modules such as CBS, Bottleneck CSP, and SPPF. The latest version employs a 6 × 6 Conv convolution module and a serial SPPF structure, which reduces computational cost while improving detection speed. The neck network adopts the FPN+PAN structure, where FPN transfers high-level semantic information to low-level layers, enhancing classification capabilities, while PAN complements positional information from bottom to top. The prediction end relies on three feature maps of different sizes (80 × 80, 40 × 40, 20 × 20) outputted by the neck network, each dedicated to detecting small, medium, and large objects. The entire model achieves efficient real-time performance while maintaining accuracy [32].

Addressing the issues of low pixel resolution and poor recognition effect for long-distance small targets in railway boundary intrusion scenarios, this article introduces the SE attention mechanism into the neck network to strengthen the correlation between channel map image feature information, thereby improving the accuracy of target recognition. Secondly, a prediction layer is added to the original network to detect long-distance small targets. Finally, by improving the anchor box and boundary loss function, the recognition accuracy and positioning precision are enhanced. The overall network architecture diagram of the improved YOLOv5s railway track foreign object detection is shown in Figure 8.

#### 2.2.2. SE Module Attention Mechanism

To strengthen the connection between channels and image features and enable effective modeling, the SE module utilizes a feature recalibration technique to learn the weight of each channel [33]. As illustrated in Figure 9, it consists of three parts: Squeeze operation, Excitation operation, and Scale operation. The Squeeze operation globally averages the input feature map, reducing the dimensionality of each channel’s feature values to a global vector, aiming to capture the global information of each channel. The Excitation operation, comprising two fully connected layers, a ReLU activation function, and a Softmax activation function, first reduces and then increases the dimensionality, ultimately generating a weight vector through the sigmoid function, ensuring that their sum is 1. The Scale operation multiplies the channel attention weights obtained in the previous step by the original input feature map. The SE module attention mechanism allows the model to focus more on the channel features with the highest information content while suppressing those that are less important.

The SE module is added to the YOLOv5s’s Neck network detection layer after C3 as a separate output layer, as shown in Figure 10. By incorporating the SE module, the connection between channels and image features is established, allowing global information to be fully utilized. This gives higher weights to small-object channels, better fitting relevant feature information and enhancing the model’s performance in small-object recognition.

#### 2.2.3. Add a Prediction Layer

The YOLOv5s model possesses three prediction layers capable of processing 640 × 640 images. Its Neck network generates 80 × 80, 40 × 40, and 20 × 20 feature maps through 8-fold, 16-fold, and 32-fold downsampling, respectively, used for detecting objects of different sizes. In railway clearance monitoring, identifying small objects at a long distance is a significant challenge. Therefore, an additional prediction layer is added to YOLOv5s, as shown in Figure 11. This innovation involves adding an upsampling operation to the Neck network, where after the third upsampling, the newly generated feature map is effectively fused with the second layer of the backbone network, resulting in a 160 × 160 feature map specifically designed for detecting small objects. The improved model boasts four prediction scales, leveraging not only high-resolution low-level features but also deep high-semantic information. This modification does not significantly increase complexity, ensuring the model’s real-time performance and efficiency.

#### 2.2.4. Improve the Anchor Box

Anchors are primarily used for predicting bounding boxes. YOLOv5s has predefined anchor boxes based on the COCO dataset. However, if these predefined anchor boxes are directly applied to railway perimeter intrusion monitoring, the recognition effect will be impacted. A study in [34] mentions anchor boxes generated through K-means clustering, which can effectively enhance the recognition accuracy of small targets. This paper attempted to adopt this method, but the recognition accuracy decreased. This is because the railway perimeter intrusion scenarios cover a large area and require identifying extremely small targets. When the actual size of a target differs significantly from the predefined anchor box, it can affect the recognition accuracy. Therefore, this paper employs a method of linearly scaling the anchor boxes [35], stretching them on both sides to identify small targets. The calculation formula is as follows:(6)$$x_{1}^{\prime }=\alpha x_{1}\;x_{6}^{\prime }=\beta x_{1}\;x_{i}^{\prime }=\frac{\left(x_{i}-x_{1}\right)}{\left(x_{6}-x_{i}\right)}\left(x_{6}^{\prime }-x_{1}^{\prime }\right)+x_{1}^{\prime }\;y_{i}^{\prime }=x_{i}^{\prime }\frac{y_{i}}{x_{i}}$$

In the formula, α represents the minimum scaling factor for reducing the anchor box; β represents the maximum scaling factor for enlarging the anchor box; $x_{i}$ represents the absolute horizontal coordinate of the anchor box, and $y_{i}$ represents the absolute vertical coordinate of the anchor box. By setting α=0.5, β=3, we can perform linear scaling on the anchor box, and its effect is illustrated in Figure 12.

#### 2.2.5. Improvement of the Loss Function

YOLOv5s utilizes the CIoU loss function for bounding box loss, and its calculation formula is as follows:(7)$$L_{CIOU}=1-IOU+\frac{\rho^{2}(b,b^{gt})}{C^{2}}+av$$

In it, *v* only includes the aspect ratio to be predicted. In the detection of foreign objects on railway tracks, especially for small target detection, the CIoU loss function has limited sensitivity to width and height, making it difficult to accurately reflect the actual status of the target, especially when dealing with objects with abnormal aspect ratios or irregular shapes. The lack of adaptive adjustment capabilities for multi-sample differences affects the convergence speed.

To address this issue, this paper introduces the Inner-IoU loss function for auxiliary bounding boxes [36], which optimizes detection performance by adjusting the scale of the auxiliary bounding boxes. Smaller-scale bounding boxes accelerate the regression of high-IoU samples, while larger-scale bounding boxes accelerate the regression of low-IoU samples. Meanwhile, a scale factor “ratio” is introduced to achieve flexible adjustment of the bounding box scale, thus obtaining regression results quickly and efficiently. As shown in Figure 13, the illustrations represent the Inner-IoU calculation methods for smaller and larger scales. The calculation method for applying Inner-IoU to the CIoU loss function is as follows:(8)$$b_{l}^{gt}=x_{c}^{gt}-\frac{w^{gt}\times ratio}{2},b_{b}^{gt}=x_{c}^{gt}+\frac{w^{gt}\times ratio}{2}$$
(9)$$b_{t}^{gt}=y_{c}^{gt}-\frac{h^{gt}\times ratio}{2},b_{b}^{gt}=y_{c}^{gt}+\frac{h^{gt}\times ratio}{2}$$
(10)$$b_{l}=x_{c}-\frac{w\times ratio}{2},b_{r}=x_{c}+\frac{w\times ratio}{2}$$
(11)$$b_{t}=y_{c}-\frac{h\times ratio}{2},b_{b}=y_{c}+\frac{h\times ratio}{2}$$
(12)$$\mathrm{int}er=(\min(b_{r}^{gt},b_{r})-\max(b_{l}^{gt},b_{t})\times (\min(b_{b}^{gt},b_{b})-\max(b_{t}^{gt},b_{t})$$
(13)$$union=(w^{gt}\times h^{gt})\times (ratio{)}^{2}+(w\times h)\times (ratio{)}^{2}-\mathrm{int}er$$
(14)$$IoU^{inner}=\frac{\mathrm{int}er}{union}$$

## 3. Experiments and Results

### 3.1. Dataset and Experimental Environment

#### 3.1.1. Foreign Object Invasion Dataset

The establishment of the sample library utilizes both railway field collection and simulated intrusion collection. Due to the limited number of intrusion cases, the primary method is simulated intrusion. By simulating typical intrusion scenarios in the Guangzhou EMU Test Site and real railway lines, a sample library of three common types of foreign object invasions has been established. The sample categories include vehicles, pedestrians, and falling rocks, totaling 2534 images. These images are divided into daytime application datasets and nighttime application datasets based on different lighting conditions, and extreme weather application datasets based on different weather conditions, including heavy fog, rain, snow, and other extreme weather. The constructed dataset is shown in Table 1.

The dataset was annotated using the LabelImg v1.8.1 open-source tool, a commonly used deep learning image annotation software. Each obstacle in the images was labeled with the smallest possible rectangular box, and the obstacle’s name was used as the target category. The number and size of labeled boxes for each sample are shown in Table 2, where small target boxes account for 75.10%. The foreign object detection dataset adopts the VOC format and is divided into training, validation, and test sets in a ratio of 8:1:1.

#### 3.1.2. Railway Track Division

In the research field of railway track division, there is currently no standardized dedicated dataset. Therefore, this paper establishes a dataset with diverse detection areas to meet the needs of image segmentation tasks. The scene images for the detection areas originate from online sources and real-world photographs, totaling 1000 images. During the annotation process, only the track detection area is considered as the positive sample category, while the rest are classified as background.

#### 3.1.3. Training Configurations and Evaluation Metrics

The host configuration used in the experiment is as follows: The CPU is an Intel(R) Core (TM) i7-10700, and the GPU is a GeForce RTX3050 with 8 GB of dedicated memory. The software configuration is as follows: The operating system is Windows 11, CUDA version 11.3, the programming language is Python 3.8, and the deep learning framework is Pytorch 1.12.1. The number of training iterations is set to 300, and the input image size is 640 × 640. For the semantic segmentation task, the MIoU metric is adopted, where a higher value indicates better segmentation performance of the model [37,38]. The calculation formula is below:(15)$$A_{MIoU}=\frac{1}{q}\sum_{u=1}\frac{p_{uu}}{\sum_{v=1}^{q}p_{uv}+\sum_{v=1}^{q}p_{vu}-p_{uu}}$$

In the formula, $A_{MIoU}$ represents the MIoU; q represents the number of label categories; $p_{uu}$ represents the number of pixels whose label category is u and that are actually predicted as u; $p_{uv}$ represents the number of pixels whose label category is v but that are actually predicted as u; and $p_{vu}$ represents the number of pixels whose label category is u but are actually predicted as v.

For the task of foreign object intrusion detection on railway tracks, there are two labels in the images: background and object. The prediction boxes are also classified as correct or incorrect, resulting in four types of samples during evaluation: $T_{p}$ (true positive), $F_{P}$ (false positive), $T_{N}$(true negative), and $F_{N}$ (false negative). These represent the cases where an actual intrusion is detected as an intrusion, an actual non-intrusion is detected as an intrusion, an actual non-intrusion is correctly detected as a non-intrusion, and an actual intrusion is falsely detected as a non-intrusion, respectively. Based on these samples, the precision (P) and recall of the model can be calculated. The formulas for these calculations are below:(16)$$P=\frac{T_{p}}{T_{p}+F_{p}}$$
(17)$$R=\frac{T_{p}}{T_{p}+F_{N}}$$

The *mAP* metric is used to measure the detection accuracy for each category. When iterating through each predicted bounding box in the test images, the corresponding precision (*P*) and recall (*R*) are calculated, and a *P–R* curve is plotted. The area under the *P–R* curve is defined as the average precision (AP). The *mAP* is obtained by calculating the average of *AP* for all categories. Typically, the *mAP* value is calculated at a confidence threshold of 0.5, denoted as *mAP*@0.5. The mean average precision reflects both precision and recall, capturing both false positive and false negative rates [39]. Therefore, *mAP*@0.5 is adopted to evaluate the recognition performance of the model.
(18)$$A_{p}=\int_{0}^{1}PdR$$
(19)$$mAP=\frac{\sum_{n=1}^{N}A_{p}(n)}{N}$$

### 3.2. Segmentation Results of Railway Tracks

The results of railway track segmentation combined with target detection can be used to determine whether there are foreign objects within the track boundary area. To accurately achieve the segmentation of the track area, the labeled detection area data are trained according to the method in Figure 3, and the training curve is shown in Figure 14. As can be seen from Figure 14, for the self-built segmentation dataset, the improved UNet segmentation model has completed the training well, with an $A_{MIoU}$ value reaching 91.8%, representing a 3.9% increase compared to before the improvement.

As can be seen from Figure 15, the improved algorithm in this paper extracts the track edges with clearer outlines and better integrity.

### 3.3. Analysis of the Results for Model Performance

#### 3.3.1. Comparative Analysis of Attention Mechanisms

To further validate the performance of the improved algorithm proposed in this experiment, multiple comparative ablation tests were conducted during the experimental process. Firstly, in terms of the choice of attention mechanisms, several common attention modules were compared, including the CA attention module, CBMA attention module, CBMAC3 attention module, Shuffle module, and SE attention module used in this paper. After 300 training epochs, inference validation was performed on the same test set. The final practical results indicate that, in this experiment, the SE attention module achieved the best recognition performance. The recognition accuracy on the validation set is shown in Figure 16.

The comparison results are shown in Table 3. Only YOLOv5s-CA experienced a decrease of 1.5% compared to the original YOLOv5s. All other models achieved a certain level of improvement in accuracy after adding attention mechanisms, among which YOLOv5s-SE had the highest increase in precision, improving by 2.2%. Therefore, this paper demonstrates that adding the SE module after the C3 detection layer of the Neck network leads to better detection performance.

#### 3.3.2. Comparative Analysis of Anchor Boxes

Based on the YOLOv5s-SE model, the recognition accuracy for small objects was improved by modifying the anchor boxes. A comparative analysis was conducted between the anchor boxes generated by K-means clustering as described in reference [35] and the anchor boxes produced by the linear scaling method outlined in this paper. After 300 training epochs, inference validation was performed on the same test set, as shown in Figure 17. The final results indicate that the method from reference [35] led to a decrease in accuracy, while the method used in this paper resulted in a more significant improvement in accuracy.

#### 3.3.3. Comparative Analysis of Model Recognition Accuracy

To verify the impact of various improvement methods on the detection of foreign object encroachment on railways, five sets of experiments were designed to evaluate each method. The Figure 18 shows the impact of different methods on the encroachment detection. Among them, Optimization 1 adds an attention mechanism, Optimization 2 adds a prediction layer based on Optimization 1, Optimization 3 improves the anchor box using offline scaling based on Optimization 2, and Optimization 4 modifies the loss function based on Optimization 3.

In Table 4, the sequential introduction of the four optimization methods continuously improves the mean average precision (*mAP*) and detection speed in frames per second (FPS). Ultimately, the improved model achieves a 7.4% increase in *mAP* and a 26.8 FPS enhancement in detection speed. Among these optimizations, Optimization 2, which involves adding a new prediction layer, has the most significant impact on recognition performance, resulting in a 3.5% increase. This is because Optimization 2 can generate larger feature maps to predict smaller sizes, making it easier to identify long-distance intruders. Additionally, Optimization 3 has the most significant improvement in detection speed, increasing it by 16.2 FPS. Therefore, the method presented in this paper can maintain the real-time performance of the algorithm while achieving higher detection accuracy.

The four optimization methods proposed in this paper are S (SE attention mechanism), P (prediction layer), B (modifying the anchor box), and L (loss function). To verify the effectiveness of these four different improvement methods, ablation experiments were designed from the following two aspects: (1) based on the original YOLOv5s algorithm, each improvement method was added individually to validate the improvement effect of each method on the original algorithm; (2) based on the YOLOv5s algorithm, each improvement method was eliminated individually to verify the degree of influence each method has on the final algorithm.

The “√” symbol indicates the introduction of the method. As can be seen from Table 5, compared to the original YOLOv5s algorithm, the introduction of each module led to a certain improvement in detection accuracy, with the newly added prediction layer showing the most significant improvement in detection accuracy, reaching 4.4%, but with a certain decrease in detection speed. The modification of the anchor box resulted in the most significant improvement in detection speed, reaching 39.3 FPS. However, compared to the final YOLOv5s-SPBL algorithm, eliminating the prediction layer had the greatest impact on detection accuracy, reaching 4.9%, while eliminating the loss function had the smallest impact on detection speed, with only a decrease of 1.6 FPS.

Figure 19 shows the visualization of some recognition results, with the original YOLOv5s recognition image on the left and the improved YOLOv5s recognition image on the right. Figure 19a is an image of foreign object encroachment recognition at a long distance (200 m from the camera) on a cloudy day. The improved model identifies a target that was falsely detected by the original model. Figure 19b is an image of foreign object encroachment recognition at a relatively long distance (200 m from the camera) during the daytime. The improved model detects a target that was missed by the original model. Figure 19c is an image of foreign object recognition at a long distance (150 m from the camera) at night. The improved model identifies a target that was falsely detected by the original model. Figure 19d is an image of foreign object recognition at a relatively long distance (100 m from the camera) at night. The improved model detects a target that was missed by the original model. Figure 19e shows the recognition situation during the daytime at a long distance (200 m from the camera). The improved model has a higher confidence score for identifying the invading target. From the comparison of multiple scenarios, it can be seen that the improved YOLOv5s model has improvements in confidence score, false detection, missed detection, and other aspects for the recognition of long-distance targets of railway foreign object encroachment.

To further validate the effectiveness and superiority of the improved model, the proposed algorithm in this paper is compared with current mainstream algorithmic models through contrast experiments under the same scenario. The experimental comparison is conducted with Faster R-CNN, SSD, YOLOv3, YOLOv7, YOLOv8, YOLOv8s-tiny, YOLOv5s, and YOLOv5s-ghostNet, and the results are shown in Table 6. Compared with other mainstream detection models, the proposed algorithm maintains a high detection speed and detection accuracy. It has a significant advantage in detection speed over the traditional two-stage algorithm Faster R-CNN, with a detection speed increase of 54.7 FPS and a mean average precision improvement of 6.4%. Compared with algorithms such as SSD, YOLOv3, YOLOv7, YOLOv8s-tiny, and YOLOv5s-ghostNet, the improved model achieves a certain improvement in accuracy while maintaining a good detection speed, demonstrating its outstanding overall performance and proving the feasibility and superiority of this algorithm.

#### 3.3.4. Analysis of Bounding Box Localization Accuracy

The detection box format output by the YOLOv5s detection method is (*x, y, w, h*), representing the center coordinates and width and height of the detection box. A comparative analysis of the localization accuracy of the detection boxes in Figure 19e is presented in Table 7. Comparing YOLOv5s with the method in this paper, it can be seen that the proposed method achieves a higher intersection over union (IoU) value in terms of localization box accuracy, indicating that the proposed method is more precise in object localization.

### 3.4. Judgment of Foreign Object Encroachment Behavior

For the detection of foreign object encroachment, it is crucial to determine whether a foreign object has entered the restricted area for the safe operation of trains. Therefore, by combining the detection results of foreign object encroachment with the restricted area, a judgment of the encroachment behavior is made. The encroachment model is shown in Figure 7. If the horizontal coordinate value of the center point of the detection box is less than the left boundary value of the restricted area, the right-upper and right-lower vertices of the detection box are judged for encroachment by calculating the restricted area coordinates. If the conditions are met, it can be determined that the right boundary of the foreign object encroaches. Conversely, it can be determined that the detected foreign object does not encroach. Similarly, it can be judged whether the left boundary of the foreign object encroaches. A judgment analysis of the encroachment behavior for Figure 19b is conducted, and the analysis results are shown in Table 8.

As shown in the table above, the proposed judgment method in this section accurately determines the encroachment behavior of the detected objects in the image. Moreover, it only requires the evaluation of specific feature points of the detection box, resulting in relatively low computational complexity and demonstrating practical usefulness.

## 4. Conclusions

As one of the greatest potential hazards threatening the safety of train operations, foreign object encroachment on railway tracks has been studied in order to improve the detection of foreign object encroachment at railway perimeters and enhance the generalization ability of detection models in small object scenarios. Based on deep learning, a method for detecting foreign object encroachment on railway tracks has been investigated. The following conclusions have been drawn:By utilizing an improved UNet semantic segmentation network and the least squares method, a linear equation for the railway track is determined, and a track boundary model is established. Through a self-constructed segmentation dataset, the improved UNet segmentation model achieved a value of 91.8%, representing a 3.9% improvement compared to the previous version. The extracted track edges are clearer and more complete.The improved YOLOv5s foreign object detection model incorporates the SE attention mechanism, adds a prediction layer, and modifies the anchor boxes and loss function. It achieves feature weighting processing with relatively low computational cost, and the detection accuracy of the model is improved by 7.4%. The proposed detection model outperforms Faster R-CNN, SSD, YOLOv3, and YOLOv8s in terms of average precision for small object detection, enabling long-distance foreign object detection.The proposed combination of the railway boundary model and the object detection model provides an effective solution for the identification and localization of foreign object encroachment on railway tracks. Through experimental validation, this method can accurately determine foreign object encroachment behaviors. Moreover, the model only requires evaluation of specific feature points of the detection box, resulting in relatively low computational complexity and demonstrating practical usefulness. It can provide strong technical support for railway transportation safety.Further work could involve collecting a more comprehensive and detailed dataset of different foreign object types, such as animals, as well as acquiring data from special environments with low illumination and low resolution to improve recognition accuracy. Additionally, exploring the integration of lidar or infrared images with video images to enhance foreign object recognition accuracy in special environments is a promising direction.

## Figures and Tables

**Figure 1 sensors-24-04483-f001:**
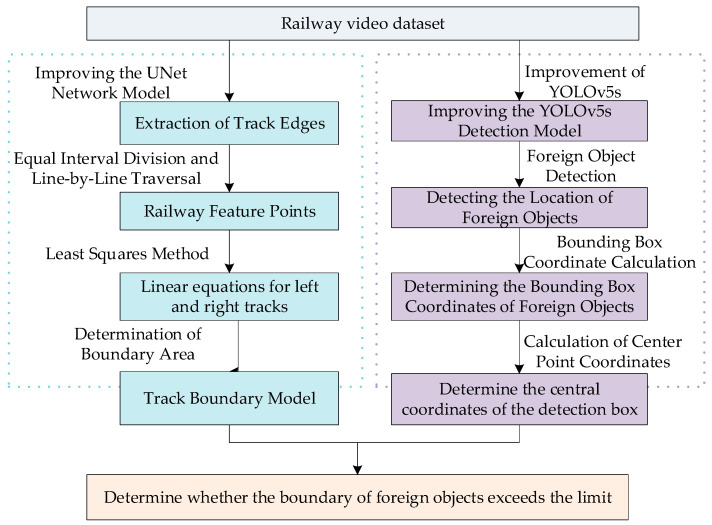
Flowchart for detecting foreign objects entering the limited area.

**Figure 2 sensors-24-04483-f002:**
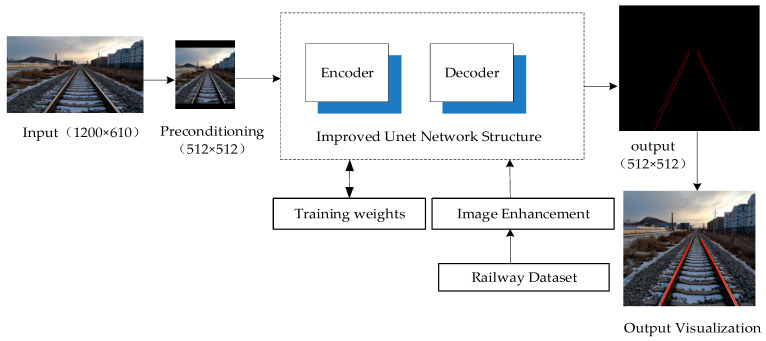
Overall flowchart of the track segmentation method.

**Figure 3 sensors-24-04483-f003:**
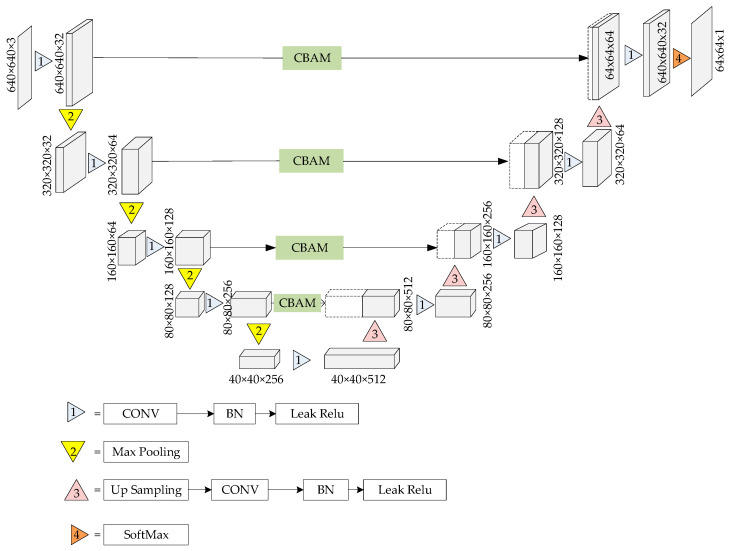
Depicts the improved UNet network structure.

**Figure 4 sensors-24-04483-f004:**
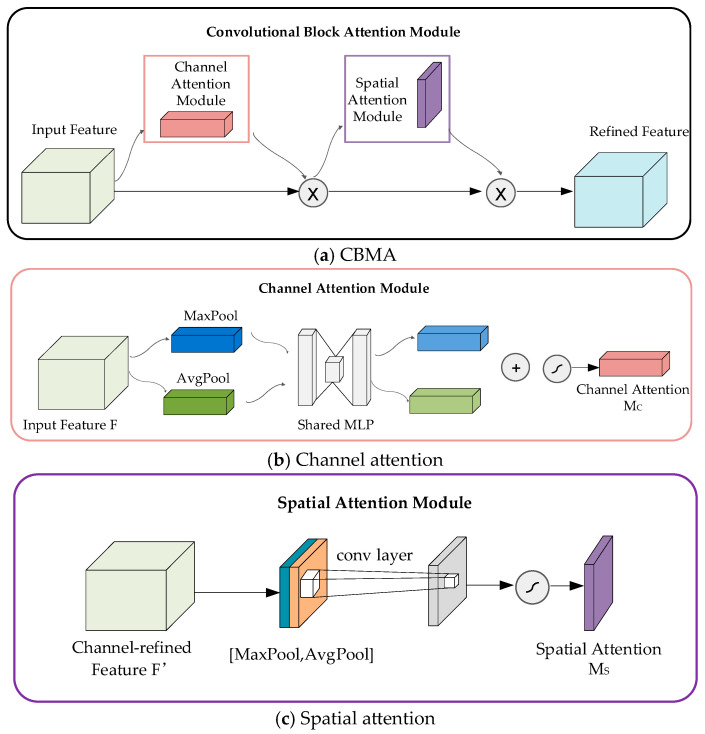
Structure diagram of the CBAM module.

**Figure 5 sensors-24-04483-f005:**
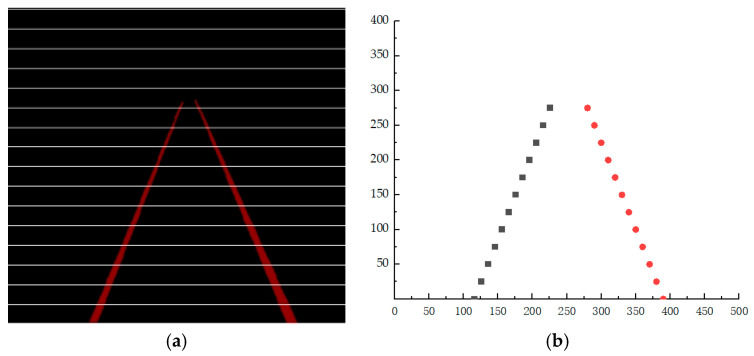
Feature extraction diagram. (**a**) Track equal division diagram. (**b**) Track feature points.

**Figure 6 sensors-24-04483-f006:**
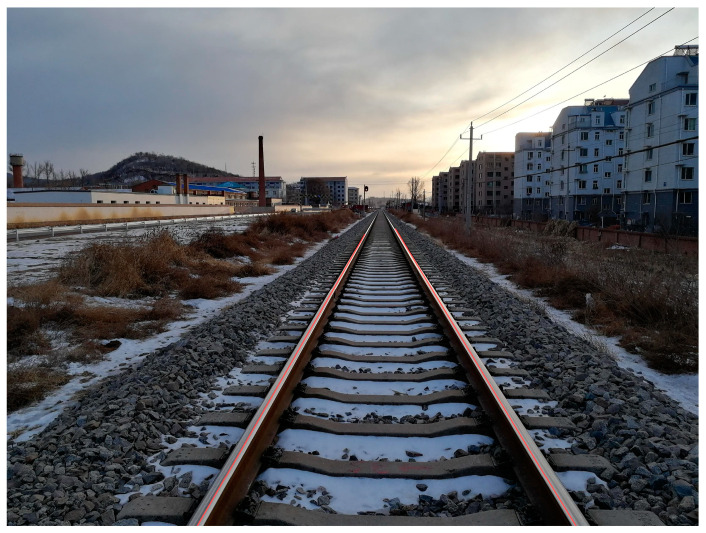
Track-fitting result diagram.

**Figure 7 sensors-24-04483-f007:**
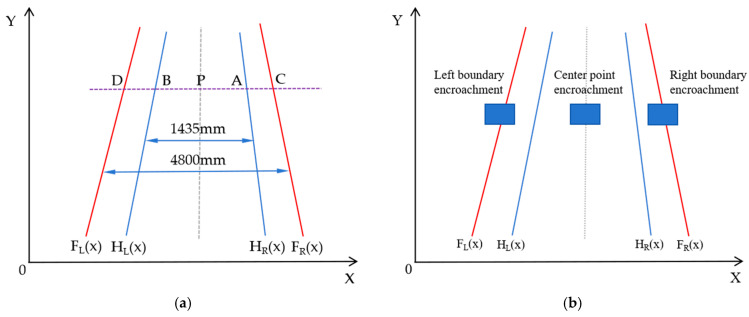
Track boundary model. (**a**) Track boundary mathematical model. (**b**) Schematic diagram of foreign object intrusion modes.

**Figure 8 sensors-24-04483-f008:**
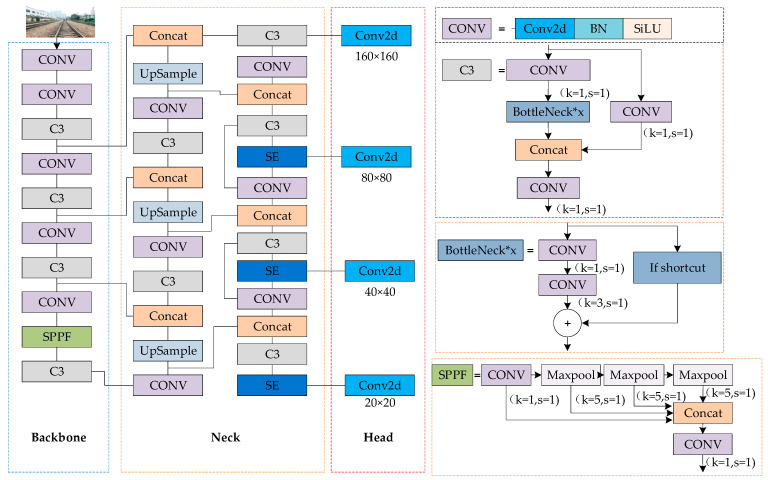
Overall network architecture of the improved YOLOv5S.

**Figure 9 sensors-24-04483-f009:**
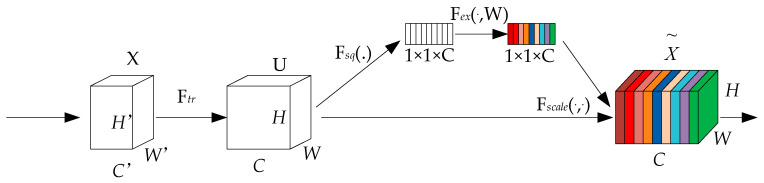
Diagram of SE module.

**Figure 10 sensors-24-04483-f010:**
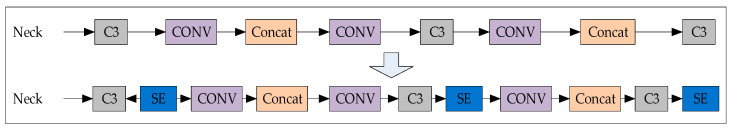
Partial Network Structure of YOLOv5s after adding SE.

**Figure 11 sensors-24-04483-f011:**
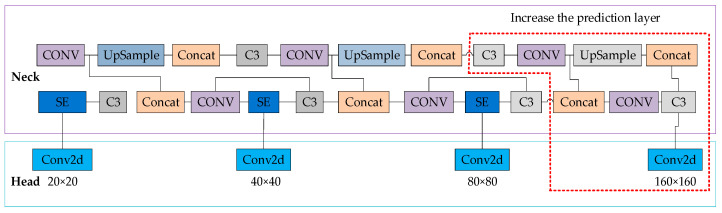
Network structure diagram with an added prediction layer.

**Figure 12 sensors-24-04483-f012:**
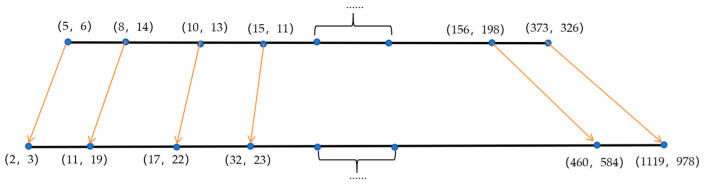
Anchor box-scaling effect diagram.

**Figure 13 sensors-24-04483-f013:**
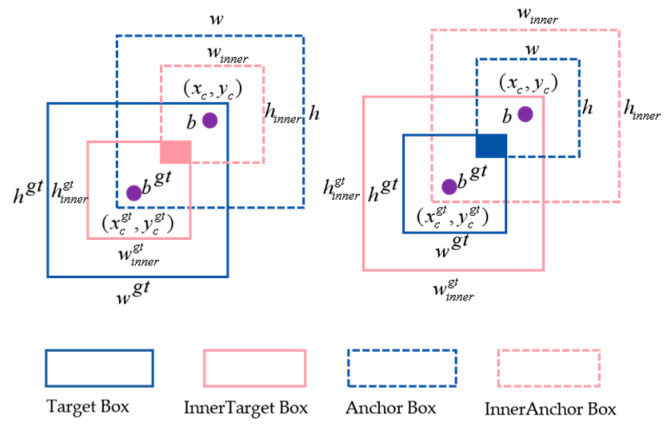
Schematic diagram of Inner-IoU.

**Figure 14 sensors-24-04483-f014:**
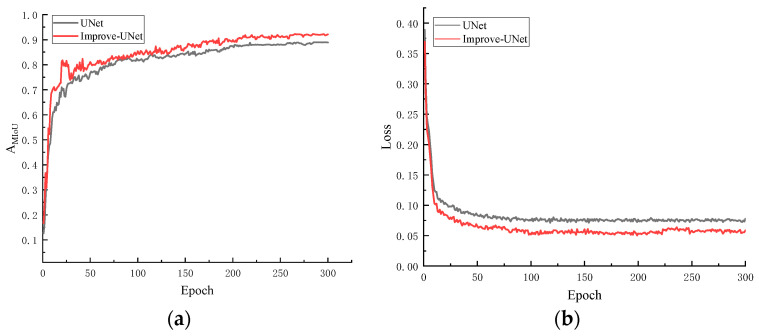
Model training results. (**a**) The accuracy of the model during the training process; (**b**) the loss value during the model training process.

**Figure 15 sensors-24-04483-f015:**
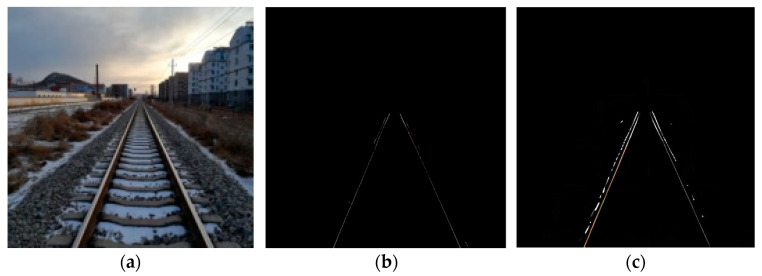
Comparison of the algorithm in this paper with the experimental results of UNet. (**a**) The original image; (**b**) UNet; and (**c**) the method in this paper.

**Figure 16 sensors-24-04483-f016:**
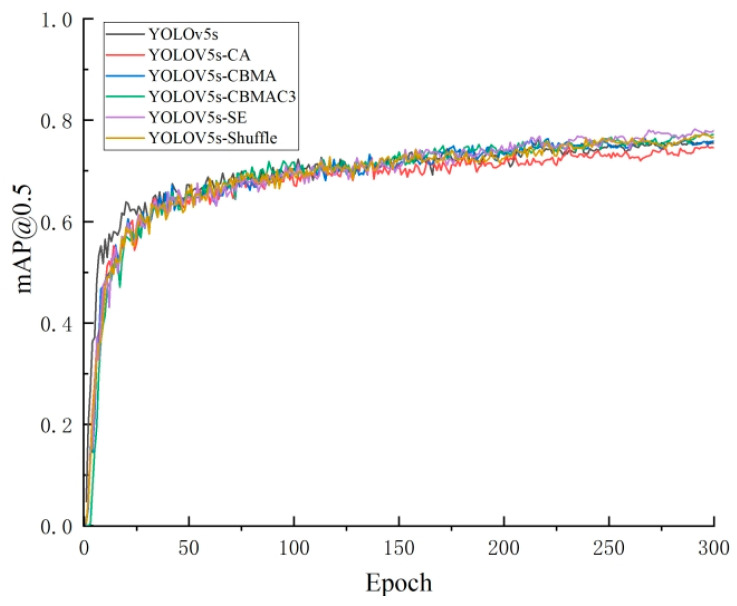
Comparison of *mAP*@0.5 during the training process with different attention mechanisms.

**Figure 17 sensors-24-04483-f017:**
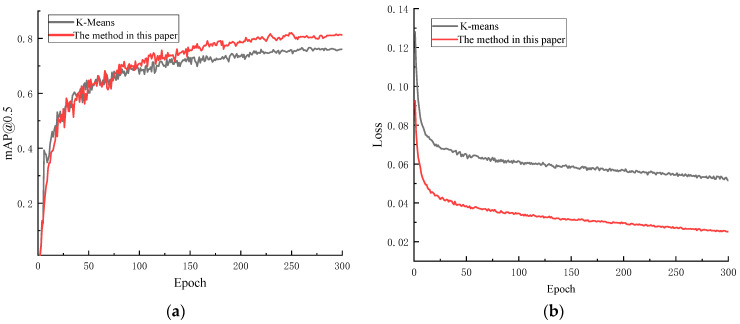
Model training results. (**a**) Comparison chart of *mAP*@0.5 during model training process; (**b**) loss values during model training process.

**Figure 18 sensors-24-04483-f018:**
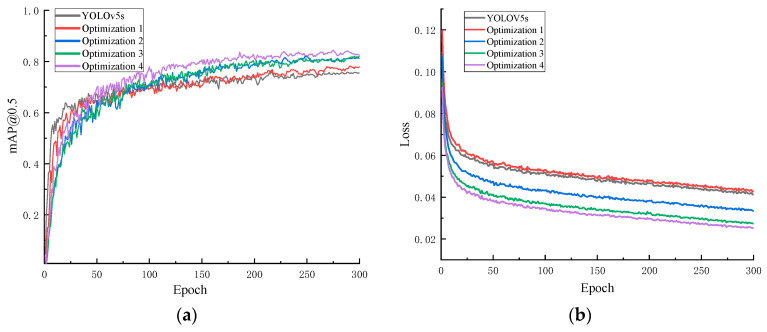
Model training results. (**a**) *mAP*@0.5 of the optimized model during the training process; (**b**) loss values of the optimized model during the training process.

**Figure 19 sensors-24-04483-f019:**
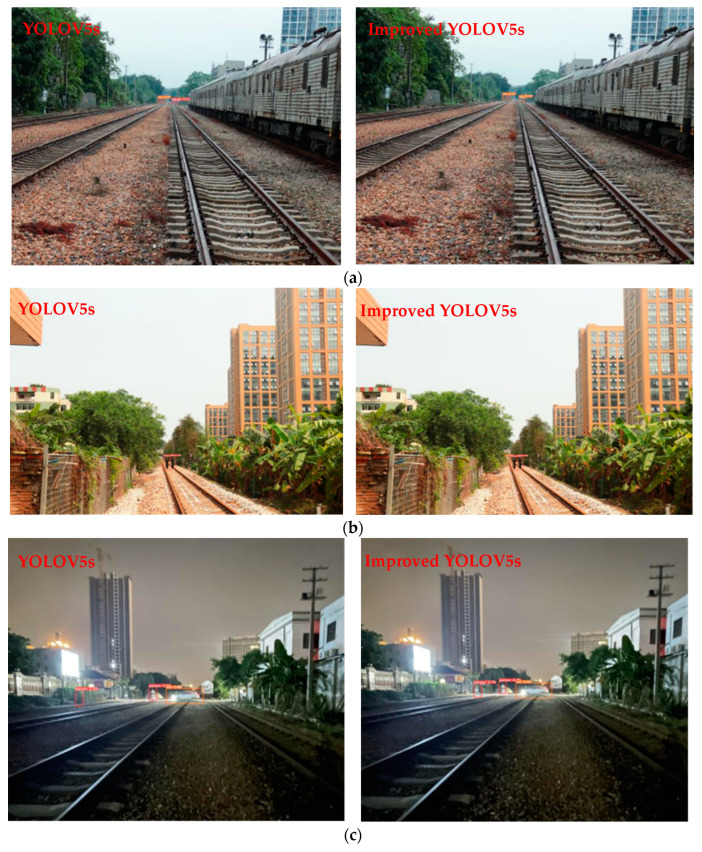
Comparison of recognition effects between two models. (**a**) False detection at a long distance on a cloudy day; (**b**) missed detection at a relatively long distance during the daytime; (**c**) false detection at a long distance at night; (**d**) missed detection at a relatively long distance at night; and (**e**) recognition at a long distance during the daytime.

**Table 1 sensors-24-04483-t001:** Overview of dataset samples.

Name of Sub-Dataset	Quantity/Images	Number of Objects
Daytime Application Dataset	1056	3852
Nighttime Application Dataset	896	2736
Extreme Weather Application Dataset	582	1116

**Table 2 sensors-24-04483-t002:** Distribution of annotated bounding boxes in the foreign object dataset.

Bounding Box Size	Quantity	Proportion
[0, 50]	4517	75.10%
[50, 100]	693	11.52%
[100, 200]	457	7.60%
[200, 300]	227	3.77%
[300, 400]	113	1.88%
[400, 800]	8	0.13%

**Table 3 sensors-24-04483-t003:** Comparison of model performance results.

	*mAP*@0.5 (%)	Precision (%)	Recall (%)	FLOPs/G
YOLOv5s	75.7	81.7	72.5	15.8
YOLOv5s-CA	74.2	85.9	69.7	15.8
YOLOv5s-CBMA	76.0	81.5	72.5	15.9
YOLOv5s-CBMAC3	77.2	88.3	72.7	22.6
YOLOv5s-SE	77.9	89.3	70.8	15.8
YOLOv5s-Shuffle	77.1	88.2	72.6	15.8

**Table 4 sensors-24-04483-t004:** Comparison of *mAP*@0.5 for each optimization method.

Experimental Group	SE	Adding a Prediction Layer	Modifying the Anchor Box	Loss Function	*mAP*@0.5 (%)	Precision (%)	Recall (%)	FPS
YOLOv5s					75.7	81.7	72.5	84.3
Optimization 1	√				77.9	84.3	70.8	90.9
Optimization 2	√	√			81.4	82.7	74.9	93.3
Optimization 3	√	√	√		82.0	87.0	75.3	109.5
Optimization 4	√	√	√	√	83.1	89.3	76.0	111.1

**Table 5 sensors-24-04483-t005:** Results of ablation experiments.

Experimental Group	SE	Adding a Prediction Layer	Modifying the Anchor Box	Loss Function	*mAP*@0.5 (%)	Precision (%)	Recall (%)	FPS
YOLOv5s					75.7	81.7	72.5	84.3
YOLOv5s-S	√				77.9	84.3	70.8	90.9
YOLOv5s-P		√			80.1	86.7	73.5	82.1
YOLOv5s-B			√		79.8	84.3	74.7	123.6
YOLOv5s-L				√	77.4	82.7	73.2	85.7
YOLOv5s-PBL		√	√	√	80.1	84.2	74.6	102.7
YOLOv5s-SBL	√		√	√	79.2	83.1	73.7	105.3
YOLOv5s-SPL	√	√		√	81.7	86.5	74.9	98.8
YOLOv5s-SPB	√	√	√		82.0	87.0	75.3	109.5
YOLOv5s-SPBL	√	√	√	√	83.1	89.3	76.0	111.1

**Table 6 sensors-24-04483-t006:** Comparison of experimental results for various models.

Model	Backbone	Input Pixel (s)	*mAP*@0.5 (%)	Precision (%)	Recall (%)	FPS
Faster R-CNN	Resnet50	1000 × 600	76.7	84.7	71.7	56.4
SSD	VGG16	300 × 300	75.4	81.3	73.4	62.1
YOLOv3	CPSDarknet53	608 × 608	74.9	80.8	71.6	82.1
YOLOv7	CPSDarknet53	640 × 640	76.6	83.1	72.9	84.7
YOLOv8s	CPSDarknet53	640 × 640	76.8	83.6	73.1	85.2
YOLOv8s-tiny	EfficientNetV3	640 × 640	77.2	85.2	73.8	94.6
YOLOv5s	CPSDarknet53	640 × 640	75.7	81.7	72.5	84.3
YOLOv5s-ghostNet	GhostNetV2	640 × 640	76.6	82.8	73.2	90.2
The method in this paper	The Algorithm	640 × 640	83.1	89.3	76.0	111.1

**Table 7 sensors-24-04483-t007:** Comparison of bounding box localization accuracy.

Detected Object	Model	Bounding Box Coordinates	Label Box Coordinates	IoU (%)
Car	YOLOv5s	(2490.5, 412, 53, 24)	(2487.5, 409, 57, 27)	72.81%
	The method in this paper	(2485, 408, 55, 28)		80.13%
Person 1	YOLOv5s	(2784.5, 434, 29, 72)	(2780, 435, 31, 74)	83.75%
	The method in this paper	(2788, 434, 32, 76)		85.07%
Person 2	YOLOv5s	(2646, 446, 34, 82)	(2647, 445, 38, 84)	80.62%
	The method in this paper	(2643.5, 447, 37, 86)		82.76%
Person 3	YOLOv5s	(2416, 443, 34, 80)	(2418, 450, 39, 91)	84.83%
	The method in this paper	(2415.5, 447, 35, 84)		86.74%

**Table 8 sensors-24-04483-t008:** Encroachment behavior judgment.

Type of Foreign Object	Vertex Coordinates	Center Coordinates	Boundary Encroachment	Type of Encroachment
Car	Upper Left (2667, 2965)	(2712.5, 2990)	Yes	Left Boundary Encroachment and Center Point Encroachment
	Lower Left (2667, 3015)		Yes	
	Upper Right (2758, 2965)		Yes	
	Lower Right (2758, 3015)		Yes	
Person 1	Upper Left (2763, 2979)	(2820.5, 3099.5)	Yes	Center Point Encroachment
	Lower Left (2763, 3220)		Yes	
	Upper Right (2878, 2979)		Yes	
	Lower Right (2878, 3220)		Yes	
Person 2	Upper Left (2876, 2968)	(2931, 3088)	Yes	Center Point Encroachment
	Lower Left (2876,3208)		Yes	
	Upper Right (2986,2968)		Yes	
	Lower Right (2986,3208)		Yes	

## Data Availability

Data are contained within the article.

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
