# Peer review of "Research on the Method of Foreign Object Detection for Railway Tracks Based on Deep Learning"

_sensors, 2024, doi:10.3390/s24144483_

Round 1
Reviewer 1 Report
Comments and Suggestions for Authors
The organization of the paper is structured. Analysis and insights on the utilized methods were presented. However, this paper suffers from weak technical detail and insufficient contribution. There is also a need to refine the experiments so as to demonstrate the advantages of the proposed method in the paper.
1. The figures in serial number 2, 3, 4, 8, 9, and 13 are not easy to observe. Please find a way to improve their clarity.
2. The paragraphs throughout the manuscript are excessively long and require rationalization. Additionally, the nomenclature within the paper needs to be harmonized, ensuring consistency (e.g., using either YOLOv5s or YOLO v5s uniformly, “Unet” or “UNet”, and “IOU” or “IoU”).
3. The algorithms used for comparison in Table 4 are too similar and need to be compared using more object detection algorithms.
4. On the line 134, Missing square brackets around the number of 30.
5. On the line 189 and 193, the symbol “*” should be changed to “×”.
6. On the line 192, The symbol Y should be the same as in Eq. (3).
7. In the figure 19, please indicate the specific names of the two models and label them clearly in the figure.
8. The abstract talks about the inadequate real-time performance of other methods, but the evaluation index about time does not appear in the experiment, please add.
9. The references style is inconsistent, please check each one.
Comments on the Quality of English LanguageThe English language is good, but it needs minor modifications.
Author Response
Dear Reviewers :
We sincerely thank reviewer for their valuable feedback that we have used to improve the quality of our manuscript. The reviewer comments are laid out below in italicized font and specific concerns have been numbered. Our response is given in normal font and changes/additions to the manuscript are given in the blue text.
Responds to the reviewer's comments:
Comments 1:[The figures in serial number 2, 3, 4, 8, 9, and 13 are not easy to observe. Please find a way to improve their clarity.]
Respones 1:[We have revised Figures in serial number 2, 3, 4, 8, 9, and 13 , enhancing their clarity, and we have also improved the clarity of other images to meet the publication requirements of the journal.]
Comments 2:[The paragraphs throughout the manuscript are excessively long and require rationalization. Additionally, the nomenclature within the paper needs to be harmonized, ensuring consistency (e.g., using either YOLOv5s or YOLO v5s uniformly, “Unet” or “UNet”, and “IOU” or “IoU”).]
Respones 2:[Thanks for your careful checks. We are sorry for our carelessness. Based on your comments, we have made the corrections to make the word harmonized within the whole manuscript. revised all "YOLO v5s" in the manuscript to "YOLOv5s", all "Unet" to "UNet", and all "IOU" to "IoU".]
Comments 3:[The algorithms used for comparison in Table 4 are too similar and need to be compared using more object detection algorithms.]
Respones 3:[Table 4 has been updated to include three new algorithms: YOLOv7, YOLOv8s-tiny, and YOLOv5s-ghostNet. At the same time, the backbone networks for all algorithms are listed, ensuring algorithm diversity, and experimental analyses have been conducted accordingly.]
Comments 4:[On the line 134, Missing square brackets around the number of 30.]
Respones 4:[Thank you very much for your careful and thorough review. I have now corrected the errors in the manuscript and conducted a detailed re-examination.]
Comments 5:[On the line 189 and 193, the symbol “*” should be changed to “×”.]
Respones 5:[ After checking the representation methods of mathematical vectors and consulting teachers in the field of mathematics, I have learned that the correct representation of vectors is indeed "×". The issue has now been corrected. Thank you for your correction.]
Comments 6:[On the line 192, The symbol Y should be the same as in Eq. (3).]
Respones 6:[We feel sorry for our carelessness. In our resubmitted manuscript, the typo is revised. Thanks for your correction.]
Comments 7:[In the figure 19, please indicate the specific names of the two models and label them clearly in the figure.]
Respones 7:[This is a great suggestion to allow readers to have a more intuitive understanding of the comparison between the two algorithms. The paper has been revised, and the specific names of the two models are clearly labeled in the figure. ]
Comments 8:[he abstract talks about the inadequate real-time performance of other methods, but the evaluation index about time does not appear in the experiment, please add.]
Respones 8:[Concerning the absence of real-time performance evaluation metrics in the experiments, we have incorporated the FPS metric to reflect the real-time capabilities of our work. In Table 4, we demonstrate the superiority of our algorithm in real-time performance by calculating the FPS of four different algorithms. Furthermore.In Table 5, through comparative analysis with current mainstream algorithms, we confirm the advantages of our algorithm in terms of detection speed and detection accuracy.]
Comments 9:[The references style is inconsistent, please check each one.]
Respones 9:[We sincerely apologize for our careless mistakes. Thank you for your reminder. After comparing with the format requirements of the journal, we found that there were inconsistencies in the format of the references and made the corresponding corrections. Thank you again for your careful review to ensure the quality of the article.]
We tried our best to improve the manuscript and made some changes marked in blue in revised paper which will not influence the content and framework of the paper. We appreciate for Reviewers’warm work earnestly, and hope the correction will meet with approval.Once again, thank you very much for your comments and suggestions.

Reviewer 2 Report
Comments and Suggestions for Authors
This manuscript introduces an advanced method for detecting foreign objects on railway tracks using deep learning techniques. The study proposes a railway boundary model created through an enhanced Unet semantic segmentation network for the precise segmentation of track categories. The authors optimize the YOLOv5s detection model by incorporating several novel enhancements, including the SE attention mechanism, a new prediction layer for small objects, a linear size scaling method for anchor boxes, and the Inner-IoU boundary regression loss function. These optimizations significantly improve detection accuracy and real-time performance. The proposed method is validated on a self-constructed dataset, demonstrating a substantial improvement in detection accuracy and robustness, particularly in complex environments.
The integration of enhanced Unet semantic segmentation with optimized YOLOv5s for railway track foreign object detection is innovative and addresses key challenges in current detection methods. The manuscript provides a thorough explanation of the enhancements made to the YOLOv5s model, such as the SE attention mechanism and Inner-IoU, which significantly improve feature extraction and detection accuracy. The use of a self-constructed image dataset for validation adds credibility to the study, and the reported improvements in MIoU and mean average precision demonstrate the effectiveness of the proposed method.
Please find some comments and suggestions below:
While the enhancements to YOLOv5s are well explained, providing more architectural details and diagrams for both the Unet and YOLOv5s models would improve clarity and understanding. Include more detailed information about the self-constructed image dataset, such as the number of images, types of foreign objects, and distribution across various conditions (e.g., different lighting or weather conditions).
Conduct a more detailed comparative analysis with other state-of-the-art models in railway track foreign object detection to highlight the advantages and potential limitations of the proposed method. Besides, it is recommended that more recent work on the application of enhanced Unet with lightweight structure for semantic segmentation, such as: https://doi.org/10.1080/17452759.2024.2325572
Although real-time performance is mentioned as a limitation of current techniques, the manuscript should provide specific metrics or benchmarks to demonstrate the real-time capabilities of the proposed method.
An ablation study to isolate the impact of each enhancement (SE attention mechanism, new prediction layer, etc.) on the overall performance would provide deeper insights into the contributions of each component.
The manuscript presents a significant advancement in the field of railway track foreign object detection using deep learning. With additional details on the model architecture, dataset, and comparative analysis, the paper has the potential to make a substantial contribution to ensuring the operational safety of rail lines. The reviewer recommendation is minor revision.
Comments on the Quality of English LanguageThe quality of English in the manuscript is generally good, with clear and coherent language.
Author Response
Dear Reviewers :
We sincerely thank reviewer for their valuable feedback that we have used to improve the quality of our manuscript. The reviewer comments are laid out below in italicized font and specific concerns have been numbered. Our response is given in normal font and changes/additions to the manuscript are given in the blue text.
Responds to the reviewer's comments:
Comments 1:[While the enhancements to YOLOv5s are well explained, providing more architectural details and diagrams for both the Unet and YOLOv5s models would improve clarity and understanding. Include more detailed information about the self-constructed image dataset, such as the number of images, types of foreign objects, and distribution across various conditions (e.g., different lighting or weather conditions).]
Respones 1:[Based on the existing work and following the reviewers' suggestions, we have provided a detailed description of the dataset. In the manuscript, we have established a sample library of three common types of foreign object invasions, including vehicles, pedestrians, and falling rocks, with a total of 2,534 images. The images are divided into daytime and nighttime application datasets based on different lighting conditions, and into extreme weather application datasets based on different weather conditions, including fog, rain, snow, and other extreme weather conditions. The specific modifications are detailed in Table 1 in the manuscript.]
Comments 2:[Conduct a more detailed comparative analysis with other state-of-the-art models in railway track foreign object detection to highlight the advantages and potential limitations of the proposed method. Besides, it is recommended that more recent work on the application of enhanced Unet with lightweight structure for semantic segmentation, such as: https://doi.org/10.1080/17452759.2024.2325572.]
Respones 2:[Thank you very much for your contribution to our research on semantic segmentation and other related areas. We have reviewed the recent work on enhanced Unet with lightweight structures for semantic segmentation, and your suggestions have provided us with more scientific and reasonable ideas and methods. We will explore this aspect in our upcoming work. ]
Comments 3:[Although real-time performance is mentioned as a limitation of current techniques, the manuscript should provide specific metrics or benchmarks to demonstrate the real-time capabilities of the proposed method.]
Respones 3:[Concerning the absence of real-time performance evaluation metrics in the experiments, we have incorporated the FPS metric to reflect the real-time capabilities of our work. In Table 3, we demonstrate the superiority of our algorithm in real-time performance by calculating the FPS of four different algorithms. Furthermore.In Table 4, through comparative analysis with current mainstream algorithms, we confirm the advantages of our algorithm in terms of detection speed and detection accuracy.]
Comments 4:[An ablation study to isolate the impact of each enhancement (SE attention mechanism, new prediction layer, etc.) on the overall performance would provide deeper insights into the contributions of each component.]
Respones 4:[The document introduces new ablation experiments designed from the following two aspects: (1) Based on the original YOLOv5s algorithm, we added only one improvement method separately to verify the effectiveness of each improvement on the original algorithm. (2) Taking the YOLOv5s algorithm as the foundation, we eliminated only one improvement method separately to validate the extent of influence each improvement has on the final algorithm. The specific experimental results are presented in Table 5 of the document. Through the ablation experiments, it is verified that the method proposed in this manuscript can maintain the real-time performance of the algorithm while achieving higher detection accuracy.]
We tried our best to improve the manuscript and made some changes marked in blue in revised paper which will not influence the content and framework of the paper. We appreciate for Reviewers’warm work earnestly, and hope the correction will meet with approval.Once again, thank you very much for your comments and suggestions.

Round 2
Reviewer 1 Report
Comments and Suggestions for Authors
The manuscript does contribute to the body of the knowledge but few suggestions need to be incorporated:
1. On the line 67, “et al” should be modified to “et al“.
2. On the line 68, 'x' should not be used to represent the multiplication sign.
3. On the line 128, “in Figure” should be changed to “in Figure 1.”.
4. In the Figure 2, the symbol “*” should be changed to “×”.
5. On the line 193, The symbol ‘a’ should be the same as in Eq. (4).
6. On the line 193, “Adding SE” should be changed to “SE module.”. And mixed used the ‘SENet’, ‘SE module’ and ‘SE Structure’, e.g., on the line 225, 242 and 256
7. On the line 371 and 372, The symbol ‘u’ and ‘v’ should be the same as in Eq. (15).
Author Response
Dear Reviewers :
We sincerely thank reviewer for their valuable feedback that we have used to improve the quality of our manuscript. The reviewer comments are laid out below in italicized font and specific concerns have been numbered. Our response is given in normal font and changes/additions to the manuscript are given in the blue text.
Responds to the reviewer's comments:
Comments 1:[On the line 67, “et al” should be modified to “et al“.]
Respones 1:[ We were really sorry for our careless mistakes. Thank you for your reminder. It has been corrected in the manuscript.]
Comments 2:[On the line 68, 'x' should not be used to represent the multiplication sign.]
Respones 2:[We are very sorry for our careless mistake. Thank you for your reminder. We have already changed "×" to "*" in the manuscript.]
Comments 3:[On the line 128, “in Figure” should be changed to “in Figure 1.”.]
Respones 3:[We are very sorry for our careless mistake. Thank you for your reminder. It has been corrected in the manuscript.]
Comments 4:[In the Figure 2, the symbol “*” should be changed to “×”.]
Respones 4:[We are very sorry for our careless mistake. Thank you for your reminder. We have already changed "*" to "×" in the manuscript..]
Comments 5:[On the line 193, The symbol ‘a’ should be the same as in Eq. (4).]
Respones 5:[We are very sorry for our careless mistake. Thank you for your reminder. It has been corrected in the manuscript.]
Comments 6:[On the line 193, “Adding SE” should be changed to “SE module.”. And mixed used the ‘SENet’, ‘SE module’ and ‘SE Structure’, e.g., on the line 225, 242 and 256]
Respones 6:[ We apologize for the imprecise expression in the manuscript. We have unified the use of "SE module" in the manuscript.]
Comments 7:[On the line 371 and 372, The symbol ‘u’ and ‘v’ should be the same as in Eq. (15).]
Respones 7:[ We were really sorry for our careless mistakes. Thank you for your reminder. It has been corrected in the manuscript.]
thanks for your careful checks. We are sorry for our carelessness. Based on your comments, we have made the corrections to make the word harmonized within the whole manuscript.We appreciate for Reviewers’warm work earnestly, and hope the correction will meet with approval.Once again, thank you very much for your comments and suggestions.
